# Cardiorespiratory Fitness and Carotid Intima–Media Thickness in Physically Active Young Adults: CHIEF Atherosclerosis Study

**DOI:** 10.3390/jcm11133653

**Published:** 2022-06-24

**Authors:** Gen-Min Lin, Pang-Yen Liu, Kun-Zhe Tsai, Yu-Kai Lin, Wei-Chun Huang, Carl J. Lavie

**Affiliations:** 1Department of Medicine, Hualien Armed Forces General Hospital, No. 100, Jinfeng St., Hualien 970, Taiwan; liupydr@gmail.com (P.-Y.L.); stupidgrandpa@yahoo.com.tw (K.-Z.T.); yukai0907@ndmctsgh.edu.tw (Y.-K.L.); 2Department of Medicine, Tri-Service General Hospital, National Defense Medical Center, Taipei 114, Taiwan; 3Department of Stomatology of Periodontology, Mackay Memorial Hospital, Taipei 104, Taiwan; 4College of Medicine, National Yang Ming Chiao Tung University, Taipei 112, Taiwan; 5Department of Critical Care Medicine, Kaohsiung Veterans General Hospital, Kaohsiung 813414, Taiwan; wchuanglulu@gmail.com; 6John Ochsner Heart and Vascular Institute, Ochsner Clinical School, The University of Queensland School of Medicine, New Orleans, LA 70121, USA; clavie@ochsner.org

**Keywords:** cardiometabolic risk factors, cardiorespiratory fitness, carotid intima–media thickness, young adults

## Abstract

**Background:** The relationship of cardiorespiratory fitness (CRF) with subclinical atherosclerosis affected by the body adiposity has been observed in children, whereas this relationship remains unclear in young adults. **Methods and Results:** A total of 1520 military recruits, aged 18–40 years, were included in Taiwan in 2018–2020. All subjects underwent detailed physical and blood laboratory examinations. CRF was evaluated by time for a 3000 m run, and subclinical atherosclerosis was evaluated by intima–media thickness of the bulb of the left common carotid artery (cIMT) utilizing high-resolution ultrasonography. Multivariable linear regression analysis with adjustments for age, sex, cigarette smoking, alcohol intake, systolic and diastolic blood pressure, high- and low-density lipoprotein cholesterols, fasting glucose, waist circumference, serum uric acid and serum triglycerides were utilized to determine the correlation between CRF and cIMT. CRF was independently correlated with cIMT (standardized β: 0.11, *p* < 0.001). Of the cardiometabolic risk markers, serum triglycerides were the only independent risk marker of cIMT (standardized β: 0.063, *p* = 0.03). In addition, the association of CRF with cIMT did not differ between those with a body mass index (BMI) ≥ 25 kg/m^2^ and those with BMI < 25 kg/m^2^ (standardized β: 0.103 and 0.117; *p* = 0.01 and 0.005, respectively). **Conclusions:** In physically active young men and women, there was an inverse association of cIMT with CRF, which was observed in both overweight/mild obesity and normal-weight individuals, highlighting the importance of endurance capacity on reducing risk of early atherosclerosis and implying that the moderation effect of body adiposity might not be present in this population.

## 1. Introduction

Atherosclerosis is a slowly progressive process starting in early life [1] and has been associated with future cardiovascular disease (CVD) events, e.g., ischemic stroke and coronary heart disease [2]. Several conventional risk factors of atherosclerosis, such as active smoking, hypertension, dyslipidemia and sedentary lifestyle, have been demonstrated in middle- and old-aged individuals [3,4,5]. In addition, taking preventive measures for these modifiable risk factors has been proven to effectively reduce the risk of CVD [6,7,8]. Thus, it is crucial to know the leading causes of subclinical atherosclerosis at young ages [9], which could potentially lead to specific measures taken early for ceasing the process of atherosclerosis and preventing the occurrence of clinical CVD. 

Carotid intima–media thickness (cIMT), measured using high-resolution B-mode ultrasonography, has been considered as a marker for subclinical atherosclerosis in children and young adults [9,10,11]. In addition, greater cIMT in young adults was strongly associated with first-time CVD events [9]. In children, obesity or overweight and intra-abdominal fat have been demonstrated to have a positive correlation with cIMT [10,12]. Additionally, cardiorespiratory fitness (CRF) was shown to have an inverse relationship for cIMT in children, which was might be mediated by waist circumference (WC) [10,13]. In contrast, higher diastolic blood pressure (BP; DBP), body mass index (BMI) and total cholesterol (TC), and lower high-density lipoprotein cholesterol (HDL-C) levels were correlated with greater cIMT in young adults in a prior meta-analysis [9]. However, it remains unclear if there is an association between CRF and subclinical atherosclerosis, such as cIMT, in young adults, and whether abdominal obesity (e.g., WC) is a significant mediator in this relationship. Therefore, this study aims to examine the association of CRF with cIMT in a large military sample of physically active young adults in Taiwan. 

## 2. Methods

### 2.1. Study Population

The study included 1822 military men and women, aged 18–40 years, without any medications for hypertension or dyslipidemia from the Cardiorespiratory Fitness and Health in Eastern Armed Forces (CHIEF) study for atherosclerosis in Taiwan in 2018–2020 [14,15,16]. All participants received daily exercise training, e.g., a 3000 m run in the morning at the military base. Each participant underwent annual physical and laboratory examinations and completed a self-report for their unhealthy behaviors, e.g., alcohol intake and cigarette smoking (active versus former or never) in the Hualien Armed Forces General Hospital. All subjects also received a high-resolution ultrasonography for a measurement of cIMT over their left carotid artery bulb after their annual health examination. CRF was evaluated by time for a 3000 m run test [17,18] carried out in the Hualien Military Physical Training and Testing Center following the annual health examination and before the end of the year [19,20,21,22]. Participants were excluded for body mass index (BMI) ≥ 30 kg/m^2^, which was an exclusion criterion for the annual run test in the military (N = 274), or serum triglycerides ≥400 mg/dL (N = 28), leaving a sample of 1520 participants for analysis. 

### 2.2. Physical and Blood Laboratory Examinations 

Each subject’s resting BP was measured once over the right arm in a sitting position through the FT201 automatic monitor device (Parama-Tech Co., Ltd., Fukuoka, Japan), utilizing the oscillometric method. The pulse pressure (PP) was defined as the difference between systolic BP (SBP) and DBP. Body height, body weight and WC were measured in a standing position. BMI was defined as body weight divided by square of body height (kg/m^2^). Overweight/mild obesity was defined as BMI 25.0–29.9 kg/m^2^ for the Taiwanese [23,24] and normal weight was defined as BMI 18.5–24.9 kg/m^2^. As for the blood tests, serum uric acid (SUA), serum triglycerides, TC, HDL-C and fasting glucose were measured by the AU640 auto analyzer (Olympus, Kobe, Japan). Each blood sample was collected following an overnight 12 h fast by experienced medical staff. Cardiometabolic risk markers were defined according to the latest International Diabetes Federation criteria for Chinese adults [25], which included fasting glucose ≥ 100 mg/dL, serum triglycerides ≥ 150 mg/dL, BP ≥ 130/85 mmHg, WC ≥ 90 cm for men or ≥80 cm for women and HDL-C < 40 mg/dL for men or <50 mg/dL for women, as well as those receiving antidiabetic, antihypertensive or lipid-lowering medications. Hyperuricemia was defined ≥7.0 mg/dL for men and ≥6.0 for women [26].

### 2.3. cIMT Measurements 

The cIMT dimension, measured by the iE33 machine (Philips Medical Systems, Andover, MA, USA) using an ultrasound scanner equipped with a linear 4–8 MHz probe, was quantitatively calculated from the leading edge of the lumen–intima interface to the leading edge of the media–adventitia interface of the left far-wall carotid artery bulb. The coefficients of variation for the repeated cIMT measurements were approximately estimated as 96.8%. For whole of the subjects, no carotid plagues were detected during the procedure of cIMT measurement.

### 2.4. Statistical Analysis 

Clinical characteristics of the participants were expressed as numbers (%) for categorical variables and mean ± standard deviation for continuous variables, respectively. Univariate linear regression was used to determine the individual correlation of CRF and each cardiometabolic risk marker with cIMT. Multiple linear regression was used to determine the association of CRF and all cardiometabolic risk markers (the BP variables in model 1 were SBP and DBP, and the BP variable in model 2 was PP) with cIMT, with adjustments for age, cigarette smoking and alcohol intake. Subgroup analyses according to sex (men and women) and the BMI level (those with overweight or mild obesity and those with normal weight) were performed. In addition, multivariable logistic regression analysis was used to determine the odds ratio (OR) of cardiometabolic abnormalities of metabolic syndrome and CRF for significant cIMT ≥ 900 μm. A value of *p* < 0.05 was considered significant. All analyses were performed utilizing SPSS version 25.0 for Windows (IBM Corp., Armonk, NY, USA). This study was approved by the Institutional Review Board of the Mennonite Christian Hospital (No. 16-05-008) in Hualien, Taiwan, and written informed consent was obtained from all participants. 

## 3. Results

### 3.1. Clinical Characteristics of the Overall Subjects

The clinical characteristics of the study population are revealed in Table 1. The mean age was 27.3 years and the mean BMI was 24.6 kg/m^2^ of overall subjects. The majority of the study population was male subjects, accounting for 88.6%. In total, there were 637 (41.9%) active smokers, 608 (40.0%) alcohol consumers, 367 participants (27.3%) with abdominal obesity, 343 overweight participants (22.6%), 317 participants (20.8%) with mild obesity and 61 participants (4.0%) fulfilled the criteria of clinically significant cIMT.

### 3.2. Correlations of Cardiometabolic Markers and CRF with cIMT in the Overall Subjects

The univariate and multivariable linear regression results of CRF and cardiometabolic risk markers with cIMT in the overall subjects are shown in Table 2. In the univariate analyses, serum triglycerides were the only cardiometabolic marker of cIMT (standardized β: 0.058, *p* = 0.03). In addition, CRF was also correlated with cIMT (standardized β: 0.113, *p* < 0.001). In the multiple linear regression models, the findings were consistent with the univariate linear regression results that both serum triglycerides and CRF were significantly correlated with cIMT in models 1 and 2 (standardized β: 0.063 and 0.11; *p* = 0.03 and < 0.001, respectively).

### 3.3. Correlations of Cardiometabolic Markers and CRF with cIMT in Men and Women

Table 3 shows the results of univariate and multiple linear regression analyses for CRF and cardiometabolic risk markers of cIMT in men and women. For men, the multiple linear regression analyses showed that serum triglycerides and CRF were significantly correlated with cIMT in models 1 and 2 (standardized β: 0.072 and 0.098; *p* = 0.02 and 0.001, respectively). There was an association with marginal significance for HDL-C in models 1 and 2 (standardized β: 0.053, both *p* = 0.08). By contrast for women, the correlation between serum triglycerides and cIMT was null, whereas the correlation of CRF with cIMT was marginally significant in models 1 and 2 (standardized β: 0.131 and 0.132; both *p* = 0.09). In addition, there was an inverse association between diastolic BP and cIMT in model 1 (standardized β: −0.207, *p* = 0.03), while there was an association with marginal significance between PP and cIMT in model 2 (standardized β: 0.152, *p* = 0.057).

### 3.4. Correlations of Cardiometabolic Markers and CRF with cIMT in Normal-Weight and Overweight Pateints

Table 4 shows the results of univariate and multiple linear regression analyses for CRF and cardiometabolic risk markers with cIMT in the normal-weight and overweight/mild obesity groups. For the overweight/mild obesity group, serum triglycerides and CRF were both significantly correlated with cIMT in models 1 and 2 (standardized β: 0.130 and 0.103; *p* = 0.003 and 0.01, respectively). In addition, there was an association between HDL-C and cIMT in models 1 and 2 (standardized β: 0.094, *p* = 0.03). For the normal-weight group, the multiple linear regression analysis shows that CRF was correlated with cIMT in models 1 and 2 (standardized β: 0.117 and 0.117; both *p* = 0.005); whereas the correlations for serum triglycerides were null in models 1 and 2. There was a borderline inverse association for DBP in model 1 (standardized β: 0.084, both *p* = 0.057). 

### 3.5. Associations of Abnormal Cardiometabolic Markers and CRF with Clinically Significant cIMT 

The Appendix A demonstrates the results of multiple logistic regression for clinically significant cIMT (≥900 µm) with abnormal cardiometabolic markers and low CRF levels in overall participants. There were no associations for any abnormal cardiometabolic markers and low CRF, possibly due to a relatively healthy and young individuals included in the present study.

## 4. Discussion

The principal findings of this study were that in physically active young adults, lower CRF and higher serum triglycerides were correlated with greater cIMT. In contrast to prior study results for children or adolescence, the association of CRF with cIMT in young adults was not moderated by the body adiposity, such as overweight or obesity. However, the association of some cardiometabolic risk markers, e.g., serum triglycerides and DBP, with cIMT might differ in the subgroup analyses according to sex and the BMI levels. 

Several prior studies have revealed that superior CRF was associated with lower risk of incident CVD and related mortality in middle- or old-aged individuals [27]. In addition, there was a report for late adolescence who had greater CRF associated with a lower risk of myocardial infarction, while the influence of obesity outweighing CRF was observed [28]. However, with regard to the effect of CRF on subclinical atherosclerosis assessed by cIMT, there were conflicting findings for children or adolescence [10,13,29,30,31]. Some reports demonstrated no association in the youth who had type 1 diabetes mellitus or overweight [10,13,29]. Ried-Larsen et al. demonstrated a sex difference in the association between physical activity levels and cIMT, which was merely observed in boys while not in girls, and the association was not modified by the adiposity [30]. Obviously, the results obtained for the present study for young adults are in line with the findings of Reid-Larsen et al., that CRF was inversely associated with cIMT regardless of the levels of BMI, and the sex difference might be due to the strength of exercise male adolescence or young adults performed greater than that the females did. 

Another crucial finding in the present study was that serum triglycerides rather than central obesity as assessed by WC were the only cardiometabolic risk marker for cIMT in the overall subjects, and specifically in men and overweight individuals. In the USE-IMT study for those aged 18–45 years [9], dyslipidemia including greater TC and lower HDL-C was associated with greater cIMT, while no information regarding the association for serum triglycerides was obtained. In addition, a meta-regression analysis revealed that those with more serum triglycerides at baseline and free of lipid-lowering therapy had a higher risk of stroke, despite no relationship between changes in serum triglycerides and the progression of cIMT [32]. As for a null association for abdominal obesity/WC shown in the present study, numerous previous studies revealed a contrary result that BMI and WC were regarded as a risk factor of greater cIMT in children or young adults [9,10]. Because our study subjects were physically active and without moderate and morbid obesity, prior studies [33,34] found that in the general population, those with overweight rather than normal weight had the lowest risk of mortality. Some of the study subjects with overweight might be related to greater muscle mass, possibly reducing the association between WC and cIMT. The present study also showed a consistent result for a null association between SUA and cIMT in asymptomatic individuals at low risk for CVD [35]. However, a controversial finding was observed in young women that there was an inverse association between DBP and cIMT, which was also observed in a few studies [36]. It is not an uncommon BP phenotype among young adults [37], characterized with isolated elevation of diastolic BP and narrow PP, which may reflect small stroke volume and has been reported with an association with lower cIMT [36]. 

### Study Strengths and Limitations

The major strength of the study was that the participants were obtained from the military wherein the exercise training program and lifestyle were similar, possibly reducing the unrecognized confounders. In addition, there were a large number of participants included in this study, which provided sufficient power to perform the subgroup analyses, despite that women accounted for merely 11.4% of the overall population. In contrast, this study had some limitations. First, because this study was carried out in a cross-sectional design, we could not approach the temporal association for CRF and cardiometabolic markers with changes in cIMT. Second, the cIMT was merely assessed over the left carotid bulb of each subject, making it possible that there could be a significant difference in the value between left and right cIMT. Third, since this study included the physically active military personnel only, the generalizability may not be applicable to the general population of young adults. Fourth, although the assessment of CRF by time for a run test has been acceptable for the youth or young adults [17,18], there is still a minor discrepancy between time for a run test and the peak or maximal oxygen uptake from the cardiopulmonary exercise test (the gold standard measure for CRF). Finally, the participants in this study were fit and homogeneous. In this case, the BMI might not be a good variable to define an “overweight” or a “mild obesity” subject, which was traditionally regarded as a poor prognostic factor of CVD in the general population.

## 5. Conclusions

Our study suggested that in physically active young men and women, there was an inverse association of cIMT with CRF, which was observed in both overweight/mild obesity and normal-weight individuals, highlighting the importance of endurance capacity to reduce atherosclerosis risk and implying that the modification effect of body adiposity in children might not be present in young adults. These results support the considerable evidence supporting the critical role of CRF in the primary and secondary prevention of CVD [38,39,40].

## Figures and Tables

**Table 1 jcm-11-03653-t001:** Clinical characteristics of the military individuals.

	N = 1520
Age, yrs	27.33 ± 5.81
Sex, male (%)	1346 (88.6)
Cigarette smoking, active (%)	637 (41.9)
Alcohol consumption, active (%)	608 (40.0)
Body mass index, kg/m^2^	24.62 ± 3.71
* Body weight category	
Normal weight (%)	860 (56.6)
Overweight (%)	343 (22.6)
Mild obesity (%)	317 (20.8)
Waist circumference, cm	82.46 ± 9.86
** Abdominal obesity (%)	367 (27.3)
Systolic blood pressure, mmHg	117.13 ± 13.20
Diastolic blood pressure, mmHg	69.27 ± 10.13
Pulse pressure, mmHg	47.86 ± 9.98
Total cholesterol, mg/dL	173.34 ± 33.32
Low-density lipoprotein cholesterol, mg/dL	106.70 ± 30.53
High-density lipoprotein cholesterol, mg/dL	50.61 ± 11.14
Serum triglycerides, mg/dL	103.22 ± 82.16
Fasting plasma glucose, mg/dL	93.31 ± 11.67
Serum uric acid, mg/dL	6.50 ± 1.47
3000 m run time, sec	891.63 ± 110.86
cIMT, mm	0.70 ± 0.13
≥0.9 mm (%)	61 (4.0)

Abbreviations: cIMT, carotid intima–media thickness. * Normal weight, overweight and mild obesity were defined as body mass index 18.5–24.9 kg/m^2^, 25.0–27.4 kg/m^2^ and 27.5–29.9 kg/m^2^ for the Taiwanese, respectively. ** Abdominal obesity was defined as waist circumference ≥ 90 cm for men and ≥80 cm for women.

**Table 2 jcm-11-03653-t002:** Correlations of cardiorespiratory fitness and cardiometabolic risk factors with carotid intima–media thickness.

	Total Cohort (N = 1520)						
		Univariate		Model 1		Model 2	
	R	Standardized β	*p*	Standardized β	*p*	Standardized β	*p*
SBP	0.060	0.016	0.55	0.034	0.35		
DBP	0.060	−0.007	0.80	−0.038	0.28		
PP	0.065	0.026	0.31			0.028	0.29
LDL-C	0.060	−0.001	0.97	−0.007	0.79	−0.007	0.79
HDL-C	0.060	0.000	0.99	0.027	0.36	0.027	0.36
TG	0.081	0.058	0.03	0.063	0.03	0.063	0.03
FPG	0.060	−0.009	0.74	−0.021	0.44	−0.021	0.44
WC	0.064	0.025	0.37	−0.004	0.88	−0.004	0.89
SUA	0.064	0.024	0.39	0.010	0.75	0.010	0.75
CRF	0.115	0.113	<0.01	0.110	<0.01	0.110	<0.01

Multiple linear regression analysis was used to determine the associations between cardiometabolic risk factors, cardiorespiratory fitness and cIMT with adjustments for age, tobacco smoking and alcohol intake. Abbreviations: CRF, cardiorespiratory fitness; DBP, diastolic blood pressure; FPG, fasting plasma glucose; HDL-C, high-density lipoprotein cholesterol; LDL-C, low-density lipoprotein cholesterol; PP, pulse pressure; SBP, systolic blood pressure; SUAs, serum triglycerides; TGs, serum triglycerides; WC, waist circumference.

**Table 3 jcm-11-03653-t003:** Correlations of physical performance and cardiometabolic risk factors with carotid intima–media thickness between men and women.

	Men (N = 1346)							Women (N = 174)						
	Univariate			Model 1		Model 2		Univariate			Model 1		Model 2	
	R	β*	*p*	β*	*p*	β*	*p*	R	β*	*p*	β*	*p*	β*	** *p* **
SBP	0.055	0.017	0.53	0.017	0.66			0.103	0.013	0.87	0.157	0.12		
DBP	0.053	0.009	0.75	−0.013	0.73			0.154	−0.117	0.13	−0.207	0.03		
PP	0.054	0.014	0.61			0.013	0.65	0.162	0.127	0.09			0.152	0.06
LDL-C	0.053	0.006	0.84	−0.003	0.92	−0.003	0.92	0.114	−0.051	0.51	−0.061	0.46	−0.056	0.50
HDL-C	0.055	0.017	0.54	0.053	0.08	0.053	0.08	0.138	−0.093	0.22	−0.103	0.19	−0.109	0.17
TG	0.079	0.061	0.03	0.072	0.02	0.072	0.02	0.111	0.043	0.57	0.029	0.74	0.021	0.80
FPG	0.053	−0.009	0.75	−0.024	0.40	−0.024	0.40	0.103	0.013	0.86	0.006	0.94	−0.008	0.92
WC	0.061	0.032	0.26	0.010	0.77	0.010	0.76	0.109	−0.039	0.61	−0.090	0.30	−0.108	0.20
SUA	0.058	0.024	0.38	0.008	0.77	0.008	0.77	0.103	0.015	0.84	0.041	0.62	0.046	0.57
CRF	0.112	0.101	<0.01	0.098	<0.01	0.098	0.001	0.153	0.113	0.13	0.131	0.09	0.132	0.09

Multiple linear regression analysis was used to determine the sex-specific associations between cardiometabolic risk factors, cardiorespiratory fitness and cIMT with adjustments for age, tobacco smoking and alcohol intake. β*: standardized β. Abbreviations: CRF, cardiorespiratory fitness; DBP, diastolic blood pressure; FPG, fasting plasma glucose; HDL-C, high-density lipoprotein cholesterol; LDL-C, low-density lipoprotein cholesterol; PP, pulse pressure; SBP, systolic blood pressure; SUAs, serum triglycerides; TGs, serum triglycerides; WC, waist circumference.

**Table 4 jcm-11-03653-t004:** Correlations of physical performance and cardiometabolic risk factors with carotid intima–media thickness between the overweight and the normal weight.

	BMI: 18.5–24.9 kg/m^2^ (N = 860)							BMI: 25.0–29.9 kg/m^2^ (N = 660)						
	Univariate			Model 1		Model 2		Univariate			Model 1		Model 2	
	R	β*	*p*	β*	*p*	β*	*p*	R	β*	*p*	β*	*p*	β*	*p*
SBP	0.082	−0.014	0.69	0.053	0.24			0.077	0.041	0.31	0.013	0.81		
DBP	0.100	−0.061	0.08	−0.084	0.06			0.077	0.043	0.30	0.008	0.88		
PP	0.089	0.040	0.25			0.052	0.14	0.067	0.011	0.79			0.010	0.81
LDL-C	0.081	−0.010	0.78	0.003	0.93	0.003	0.92	0.066	0.008	0.84	−0.005	0.91	−0.005	0.91
HDL-C	0.082	−0.017	0.64	−0.035	0.34	−0.036	0.33	0.070	0.024	0.56	0.094	0.03	0.094	0.03
TG	0.090	−0.041	0.24	−0.044	0.24	−0.047	0.21	0.127	0.113	<0.01	0.130	<0.01	0.130	<0.01
FPG	0.094	−0.050	0.15	−0.049	0.16	−0.051	0.13	0.067	0.012	0.75	−0.023	0.58	−0.023	0.58
WC	0.082	−0.019	0.61	−0.028	0.47	−0.036	0.35	0.102	0.084	0.04	0.067	0.12	0.067	0.12
SUA	0.081	0.008	0.83	0.016	0.68	0.013	0.74	0.077	0.042	0.31	0.016	0.71	0.016	0.71
CRF	0.121	0.109	<0.01	0.117	<0.01	0.119	<0.01	0.123	0.114	<0.01	0.103	0.01	0.103	0.01

Multiple linear regression analysis was used to determine the associations between cardiometabolic risk factors, cardiorespiratory fitness and cIMT in the overweight and the normal weight with adjustments for age, tobacco smoking and alcohol intake. β*: standardized β. Abbreviations: CRF, cardiorespiratory fitness; DBP, diastolic blood pressure; FPG, fasting plasma glucose; HDL-C, high-density lipoprotein cholesterol; LDL-C, low-density lipoprotein cholesterol; PP, pulse pressure; SBP, systolic blood pressure; SUAs, serum triglycerides; TGs, serum triglycerides; WC, waist circumference.

## Data Availability

The datasets generated and/or analyzed during the current study are not publicly available due to materials obtained from the military in Taiwan, which are confidential, but available from the corresponding author on reasonable request.

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
