# Peer review of "Cardiorespiratory Fitness and Carotid Intima–Media Thickness in Physically Active Young Adults: CHIEF Atherosclerosis Study"

_jcm, 2022, doi:10.3390/jcm11133653_

Round 1

Reviewer 1 Report

Purpose: this study aimed to examine the association of CRF with cIMT in a large military 61 sample of physically active young adults in Taiwan.

This was a complex paper with many different aspects to it.

Suggestions:

How can you conclude that abdominal obesity was associated or not when BMI >30 was excluded. What was the WC limitation? You are defining obesity and overweight by BMI but are excluded obese people.

What formula did you use for CRF if time was used for the 3000 meter run test. Why did you use that one?

There was a greater amount of men in the study, therefore how much can be shown about women please mention that.

Author Response

Comments and Suggestions for Authors

Purpose: this study aimed to examine the association of CRF with cIMT in a large military sample of physically active young adults in Taiwan.

This was a complex paper with many different aspects to it.

Response

Thank you very much for your kindly comments.

We have revised the manuscript as you suggested.

Suggestions:

  1. How can you conclude that abdominal obesity was associated or not when BMI >30 was excluded. What was the WC limitation? You are defining obesity and overweight by BMI but are excluded obese people.

Response

Thank you very much for this important comment.

Since the definition of body mass index (BMI) for obesity in Taiwanese (Chinese) was set as ≥27.5 kg/m2, the prevalence of those with mild obesity (BMI: 27.5-29.9 kg/m2) in our study population was estimated 20.8%, and the prevalence of abdominal obesity defined as ≥90 cm for Asian men and ≥80 cm for Asian women in our study population was estimated 27.3%.  Therefore, we can make a conclusion for no association of waist circumference with cIMT in those with normal weight and those with overweight/mild obesity.

The data of abdominal obesity, BMI defined-normal weight, overweight and mild obesity have been provided on lines 126-128 and in Table 1.

In addition, we have also added a statement “Finally, the participants in this study were fit and homogeneous. In this case, the BMI might not be a good variable to define an "overweight” or a “mild obesity” subject which was traditionally regarded as a poor prognostic factor in the general population” in the limitation section on lines 262-265.

  1. What formula did you use for CRF if time was used for the 3000 meter run test. Why did you use that one?

Response

Thank you very much for this important comment.

Time for a run test has been an acceptable measure for CRF in the youth or young adults in previous studies. Two references (ref. 17 and 18) have been added in the revised manuscript to support this critical point. In addition, we also described it in the limitation section on lines 259-262.

“Fourth, although assessment of CRF by time for a run test has been acceptable for the youth or young adults,17, 18 there still has a minor discrepancy between time for a run test and the peak or maximal oxygen uptake from cardiopulmonary exercise test (the gold standard measure for CRF).”

  1. Mayorga-Vega D, Aguilar-Soto P, Viciana J. Criterion-Related Validity of the 20-M Shuttle Run Test for Estimating Cardiorespiratory Fitness: A Meta-Analysis. J Sports Sci Med. 2015 Aug 11; 14(3):536-47.
  2. Raghuveer G, Hartz J, Lubans DR, Takken T, Wiltz JL, Mietus-Snyder M, Perak AM, Baker-Smith C, Pietris N, Edwards NM; American Heart Association Young Hearts Athero, Hypertension and Obesity in the Young Committee of the Council on Lifelong Congenital Heart Disease and Heart Health in the Young. Cardiorespiratory Fitness in Youth: An Important Marker of Health: A Scientific Statement From the American Heart Association. Circulation. 2020 Aug 18; 142(7):e101-e118.

  1. There was a greater amount of men in the study, therefore how much can be shown about women please mention that.

Response

Thank you very much for this important comment.

We have added this point in the strength and limitation section on lines 250-252.

“In addition, there were a large number of participants included in this study, which provided sufficient power to perform the subgroup analyses, despite that women accounted for merely 11.4% of the overall population.”

Reviewer 2 Report

The study sounds good, with a minor suggestion in the limitation section.

The authors need to highlight the limitation of external validity. This is considered a strength of the study by the authors, but also limits the generalization of the results.

There is a need for a better explanation of the cardiorespiratory fitness test. It does not seem to be validated against a "gold standard" but even though you should justify its use.

The sample seems very fit and homogeneous. In this case, the BMI would not be a good variable to define an "overweight" subject.

Author Response

Comments and Suggestions for Authors

The study sounds good, with a minor suggestion in the limitation section.

Response

Thank you very much for your kindly comments.

We have revised the manuscript as you suggested.

  1. The authors need to highlight the limitation of external validity. This is considered a strength of the study by the authors, but also limits the generalization of the results.

Response

Thank you very much for this important comment.

We have added this point in the limitation section on lines 257-259.

“Third, since this study included the physically active military personnel only, the generalizability may not be applicable to the general population of young adults.”

  1. There is a need for a better explanation of the cardiorespiratory fitness test. It does not seem to be validated against a "gold standard" but even though you should justify its use.

Response

Thank you very much for this important comment.

Time for a run test has been an acceptable measure for CRF in the youth or young adults in previous studies. Two references (ref. 17 and 18) have been added in the revised manuscript to support this critical point. In addition, we also described it in the limitation section on lines 259-262.

“Fourth, although assessment of CRF by time for a run test has been acceptable for the youth or young adults,17, 18 there still has a minor discrepancy between time for a run test and the peak or maximal oxygen uptake from cardiopulmonary exercise test (the gold standard measure for CRF).”

  1. Mayorga-Vega D, Aguilar-Soto P, Viciana J. Criterion-Related Validity of the 20-M Shuttle Run Test for Estimating Cardiorespiratory Fitness: A Meta-Analysis. J Sports Sci Med. 2015 Aug 11; 14(3):536-47.
  2. Raghuveer G, Hartz J, Lubans DR, Takken T, Wiltz JL, Mietus-Snyder M, Perak AM, Baker-Smith C, Pietris N, Edwards NM; American Heart Association Young Hearts Athero, Hypertension and Obesity in the Young Committee of the Council on Lifelong Congenital Heart Disease and Heart Health in the Young. Cardiorespiratory Fitness in Youth: An Important Marker of Health: A Scientific Statement From the American Heart Association. Circulation. 2020 Aug 18; 142(7):e101-e118.

  1. The sample seems very fit and homogeneous. In this case, the BMI would not be a good variable to define an "overweight" subject.

Response

Thank you very much for this important comment.

We have added a statement “Finally, the participants in this study were fit and homogeneous. In this case, the BMI might not be a good variable to define an "overweight” or a “mild obesity” subject which was traditionally regarded as a poor prognostic factor in the general population” in the limitation section as you suggested on lines 262-265.

Round 2

Reviewer 1 Report

Thank you for addressing my comments.